# Simple rules for resolved level crossing spectra in magnetic field effects on reaction yields

Dmitri V. Stass[1,2], Victor A. Bagryansky[1], Yuri N. Molin[1]

[1]Voevodsky Institute of Chemical Kinetics and Combustion, Novosibirsk, 630090, Russia
[2]Novosibirsk State University, Novosibirsk, 630090, Russia

*Correspondence to*: Dmitri V. Stass (stass@ns.kinetics.nsc.ru)

**Abstract.** In this work we derive conditions under which a level crossing line in magnetic field effect curve for a recombining radical pair will be equivalent to ESR spectrum, and discuss three simple rules for qualitative prediction of the level crossing spectra.

## 1 Introduction

Spin-correlated nature of radical (ion) pairs arising as intermediates in many natural or induced chemical transformations gives rise to a host of "magnetic and spin effects" in chemical reactions. It all started with observing (Bargon, 1967; Ward and Lawler, 1967) and understanding (Closs, 1969; Kaptein and Oosterhoff, 1969) strange-looking "polarized" NMR spectra, and has evolved into a mature field in itself with a wide range of powerful experimental and theoretical techniques relying on magnetically manipulating spins in chemical processes (Salikhov et al., 1984; Steiner and Ulrich, 1989; Hayashi, 2004), culminating in the modern high-tech finesse of advanced hyperpolarized NMR (Ivanov et al., 2014).

This paper deals with a curious bridge between the most humble magnetic field effect curves (MFE), *i.e.*, dependence of reaction yield on applied static magnetic field, and hyperpolarized NMR: additional sharp resonance-like lines that may occur against the smooth background of MFE due to genuine level crossings in the spin system of the radical pair. The lines were first discovered in zero magnetic field (Anisimov et al. 1983; Fischer, 1983) and attributed to interference of pair states in the higher, spherical, symmetry conditions of zero external field similar to Hanle effect in atomic spectroscopy (Hanle, 1924). The zero field line, or Low Field Effect, was then put to the front as the possible physical mechanism of magnetoreception, and the research that followed was plenty. However, this completely overshadowed the other, spectroscopic, aspect of the level crossing lines possible in field other than zero.

Level crossing (Dupont-Roc et al., 1969; Silvers et al., 1970; Levy, 1972; Astilean et al. 1994) and avoided crossing, or anticrossing (Eck et al., 1963; Wieder and Eck, 1967; Veeman and Van der Waals, 1970; Baranov and Romanov, 2001; Yago et al., 2007; Kothe et al., 2010; Anishchik and Ivanov, 2017; Anishchik and Ivanov, 2019), spectroscopy has long been an established tool in atomic and molecular spectroscopy, as well as solid state physics, providing structural information from specific (anti)crossing lines in nonzero fields, whose positions are determined by

interactions shaping the energy levels of the system. For radical pairs purely spin level crossings at nonzero fields in MFE first appeared in calculations in already cited paper (Anisimov et al., 1983), although they were not discussed as they were not observed in the accompanying experiments on radiolytically generated radical ion pairs. However, a year later this group published a theoretical work (Sukhenko et. al, 1985) that specifically explored level crossings in nonzero fields for radical pairs with equivalent nuclei in only one pair partner, and gave explicit expression for their position determined by the hyperfine coupling (HFC) constant. Such lines were later indeed experimentally observed in several systems by two teams (Stass et al., 1995b; Saik et al., 1995; Grigoryants et al., 1998; Kalneus et al., 2006a). Furthermore, in a subsequent paper (Tadjikov et al., 1996) it was suggested and demonstrated in numerical simulations for several systems of simple structure, and confirmed in a proof-of-principle experiment, that hyperfine structure of the second pair partner may be revealed at the level crossing lines. The earliest mentioning of the very possibility to observe a resolved structure on a level crossing line for a radical pair was probably the paper on MFE in a Ge-containing pair induced by a large difference in $g$-values of the pair partners (Shokhirev et al., 1991), where a level crossing line appeared in modeling. Later the $\Delta g-$ induced level crossing spectra were theoretically explored in detail in paper (Brocklehurst, 1999).

In this work we develop the ideas of (Sukhenko et al., 1985; Tadjikov et al., 1996; Brocklehurst, 1999) to explore how a resolved structure may appear in MFE curves containing lines due to level crossing, referred to as MARY spectra. The discussion is based on the properties of radiation-induced radical ion pairs, created by CW X-irradiation of nonpolar solutions of suitable electron donor and acceptor molecules and detected by luminescence produced by pair recombination from electron spin singlet state. To avoid a lengthy introduction to the properties of such pairs, the reader is referred to a review book chapter (Stass et al., 2011) where a detailed discussion of such pairs, as well as an introductory discussion of conventional MFE curves in terms of level (anti)crossings, can be found. For the purposes of this work it will suffice to assume that the pair starts from and recombines to a spin-correlated singlet state, its spin evolution is governed by Hamiltonian including only isotropic Zeeman and hyperfine interactions in independent pair partners, the recombination itself is not spin-selective, the relaxation can be neglected, and the theoretical counterpart to experimental observable is the Laplace transform of singlet state population $\rho_{ss}$, taken in time domain, as a function of applied static magnetic field. We shall first show analytically that for a pair containing a spin-$I$ nucleus with large HFC constant and spin $I > 1$ at one partner and a compact arbitrary hyperfine structure at the other partner a resolved ESR spectrum of the "narrow" partner is expected at the level crossing line due to the "driving" partner with large HFC, and then use this result to derive and discuss several simple rules for the possible resolved level crossing spectra.

## 2 Derivation of resolved level-crossing spectra

We start by quoting the key result of the original paper (Sukhenko et al., 1985) and recasting it in the form that is convenient for further generalization. Given a radical pair having a single spin-$I$ nucleus with HFC $a$ in only one of the partners described by Hamiltonian (setting $\hbar = 1$, $\omega_0 = g\beta B$ )

$$\hat{H} = \omega_0 \left( S_{1z} + S_{2z} \right) + a\vec{S}_1\vec{I}, \tag{1}$$

its eigenstates are divided into non-overlapping sets indexed by total anglular momentum projection $\Sigma_z = S_{1z} + S_{2z} + I_z$, and spin evolution proceeds independently in state subspaces with different values $m$ of $\Sigma_z$ with maximum dimension 4. For a pair with singlet initial state and observable recombination into singlet state the needed time-dependent probability $\rho_{ss}(t)$ is a sum of partial probabilities over subspaces

$$\rho_{ss}(t) = \sum_{m=-I}^{I} \rho_{ss}(t;m). \tag{2}$$

For a subensemble of pairs with $\Sigma_z = m, |m| < I$ the subspace includes four states with eigenvalues

$$E_1(m) = -\frac{a}{4} - \frac{\omega_0}{2} + R_m, \; E_2(m) = -\frac{a}{4} - \frac{\omega_0}{2} - R_m,$$
$$E_3(m) = -\frac{a}{4} + \frac{\omega_0}{2} + R_{m-1}, \; E_4(m) = -\frac{a}{4} + \frac{\omega_0}{2} - R_{m-1}, \tag{3}$$

where

$$2R_m = \sqrt{\omega_0^2 + a\omega_0 \left( 2m+1 \right) + a^2 \left( I + \frac{1}{2} \right)^2}. \tag{4}$$

The states with maximum possible $\Sigma_z = \pm(I+1)$, i.e., electron spin-triplet states with maximum nuclear spin projection, are isolated eigenstates and are completely excluded from pair spin evolution. For the outersmost blocks involved into spin evolution with $\Sigma_z = \pm I$ there are only three states with eigenvalues

$$E_1(m = \pm I) = \frac{aI}{2}, \; E_2(m = \pm I) = \pm\frac{\omega_0 \mp a}{2} + R, \; E_3(m = \pm I) = \pm\frac{\omega_0 \mp a}{2} - R, \tag{5}$$

where

$$2R = \sqrt{\left\{ \omega_0 \pm a \left( I - \frac{1}{2} \right) \right\}^2 + 2a^2 I}. \tag{6}$$

For each value of $m$ from the range $-I < m < I$ the levels are degenerate in pairs in zero field ($E_1 = E_3$, $E_2 = E_4$), which gives rise to the ubiquitous zero field line. In addition, for the inner blocks $|m| < I$ the levels $E_1$ and $E_4$ may become degenerate in non-zero fields as well, with crossing in the subensemble $m < 0$ for $a > 0$ and vice versa, occurring in the fields

$$\omega_0^* = -\frac{aI(I+1)}{2m}.$$ (7)

For the outermost blocks the levels become degenerate only at zero field. Thus, for a pair with a single spin-$I$ nucleus there should be a zero field line and, provided $I > 1$, additional level crossing extrema of Eq. (7) in "multiple" fields may be expected.

Picking off at this point, we take a different view at this problem. Taking advantage of results from works (Brocklehurst, 1976; Salikhov et al., 1984; Stass et al., 1995c), the sought singlet state population for an initially singlet radical pair with single spin-$I$ nucleus in arbitrary magnetic field can be written as

$$\rho_{ss}(t) = \frac{1}{4} + \frac{1}{4} p(t) + \frac{1}{2} \text{Re}\left(e^{-i\omega_0 t} h(t)\right),$$ (8)

where

$$p(t) = 1 - \frac{a^2}{2I+1} \sum_{m=-I}^{I} \frac{I(I+1) - m(m+1)}{(2R_m)^2} \left[1 - \cos(2R_m t)\right],$$ (9)

$$h(t) = \frac{1}{4(2I+1)} \sum_{m=-I}^{I} \left[(1+D_m)e^{iR_m t} + (1-D_m)e^{-iR_m t}\right]\left[(1+D_{m-1})e^{iR_{m-1}t} + (1-D_{m-1})e^{-iR_{m-1}t}\right],$$ (10)

$$D_m = \frac{\omega_0 + a\left(m + \frac{1}{2}\right)}{2R_m}$$ (11)

Assuming the simplest possible exponential recombination kinetics, the theoretical counterpart of MARY spectrum is given by Laplace transform of Eq. (8)

$$M(s, \omega_0) = \int_0^\infty e^{-st} \rho_{ss}(t) \, dt,$$ (12)

where the Laplace variable $s$ has the meaning of recombination rate, or, more generally, the inverse lifetime of the spin-correlated state of the radical pair (Stass et al., 1995a). Direct evaluation of Eq. (12) with substituted Eqs. (8-11) produces

$$sM\left(\omega_0,s\right)=\frac{1}{4}+\frac{1}{4}\left\{1-\frac{a^2}{2I+1}\sum_{m=-I}^{I}\frac{I\left(I+1\right)-m\left(m+1\right)}{\omega_0^2+a\omega_0\left(2m+1\right)+a^2\left(I+\frac{1}{2}\right)^2}\frac{1}{s^2+\left(2R_m\right)^2}\right\}$$

$$+\frac{1}{8\left(2I+1\right)}\sum_{m=-I}^{I}\left[\left(1+D_m\right)\left(1+D_{m-1}\right)\frac{s^2}{s^2+\left(R_m+R_{m-1}-\omega_0\right)^2}\right]$$

$$+\frac{1}{8\left(2I+1\right)}\sum_{m=-I}^{I}\left[\left(1+D_m\right)\left(1-D_{m-1}\right)\frac{s^2}{s^2+\left(R_m-R_{m-1}-\omega_0\right)^2}\right] \qquad (13)$$

$$+\frac{1}{8\left(2I+1\right)}\sum_{m=-I}^{I}\left[\left(1-D_m\right)\left(1+D_{m-1}\right)\frac{s^2}{s^2+\left(R_m-R_{m-1}+\omega_0\right)^2}\right]$$

$$+\frac{1}{8\left(2I+1\right)}\sum_{m=-I}^{I}\left[\left(1-D_m\right)\left(1-D_{m-1}\right)\frac{s^2}{s^2+\left(R_m+R_{m-1}+\omega_0\right)^2}\right].$$

A numerical experiment demonstrates that for positive $\omega_0$ resonance-like peaks in $M\left(\omega_0\right)$ are produced only by the terms

$$\left(1+D_m\right)\left(1+D_{m-1}\right)\frac{s^2}{s^2+\left(R_m+R_{m-1}-\omega_0\right)^2} \qquad (14)$$

at fields satisfying the condition

$$R_m+R_{m-1}-\omega_0=0, \qquad (15)$$

which is immediately seen to reproduce the level crossing condition $E_1=E_4$ of Eq. (7). All other terms in Eq. (14) produce the smoothly varying background of conventional magnetic field effect curve, related to gradual change of the eigenbasis with variation of applied magnetic field.

However, having now an explicit expression for MARY spectrum Eq. (14), we can be more quantitative in characterizing the level-crossing lines at "multiple fields" of Eq. (7). Evaluation of the prefactor $\left(1+D_m\right)\left(1+D_{m-1}\right)$ in

Eq. (14) at the crossing point of Eq. (7) produces the amplitude of the corresponding peak as

$$A\left(I,m\right)=4\frac{\left(I\left(I+1\right)-m^2\right)^2-m^2}{I^2\left(I+1\right)^2-m^2}, \qquad (16)$$

while developing Eq. (15) into Taylor series for a small deviation from the crossing point of Eq. (7) produces its Lorentzian width as

$$W\left(I,m\right)=\frac{s}{2}\left(\frac{I\left(I+1\right)}{m^2}-\frac{1}{I\left(I+1\right)}\right), \qquad (17)$$

where the Laplace variable $s = \tau^{-1}$ is the inverse of the genuine exponential lifetime of the pair.

The formalism of Eq. (8) makes it very convenient to introduce a spin-$I_2$ nucleus at the other partner of the pair. The corresponding counterpart to Eq. (8) would read

$$\rho_{ss}(t) = \frac{1}{4} + \frac{1}{4}p_1(t)p_2(t) + \frac{1}{2}\text{Re}\left(h_1(t)h_2^*(t)\right), \tag{18}$$

where subscripts 1 and 2 relate the corresponding functions in Eqs. (9,10) to the 1st and 2nd pair partner, with their respective

nuclear spins $I_{1,2}$ and coupling constants $a_{1,2}$ introduced as appropriate, and "*" stands for complex conjugation. The summations in functions of Eqs. (9,10) run over all $2I_{1,2} + 1$ values of the respective nuclear spin projections.

The last term in Eq. (18) containing $\text{Re}\left(h_1(t)h_2^*(t)\right)$ now produces for each pair $(m,n)$ 16 terms in $\rho_{ss}$ of the form

$$\left(1 \pm D_{1,m}\right)\left(1 \pm D_{1,m-1}\right)\left(1 \pm D_{2,n}\right)\left(1 \pm D_{2,n-1}\right)\exp\left[i\left(\pm R_{1,m} \pm R_{1,m-1} \mp R_{2,n} \mp R_{2,n-1}\right)t\right], \tag{19}$$

and again numerical experiment demonstrates that for positive fields the only resonance-like contributions to the Laplace transform $M(\omega_0)$ come from the terms

$$\left(1 + D_{1,m}\right)\left(1 + D_{1,m-1}\right)\left(1 + D_{2,n}\right)\left(1 + D_{2,n-1}\right)\frac{s^2}{s^2 + \left(R_{1,m} + R_{1,m-1} - R_{2,n} - R_{2,n-1}\right)^2}, \tag{20}$$

with positions of the maxima determined by equation

$$R_{1,m} + R_{1,m-1} - R_{2,n} - R_{2,n-1} = 0, \tag{21}$$

while all other terms only contribute to the smooth background.

Equation (21) is equivalent to 8th order algebraic equation and does not lend itself to exact analytic solution. To advance further, we shall now impose the assumption that $a_2 \ll a_1$ and focus on the vicinity of one of the crossing points of Eq. (7) for the "dominant" partner with the larger HFC. Proceeding in two steps now, we first note that these assumptions automatically place the second partner in the high field limit $a_2 \ll \omega_0$, which lets develop the square roots $R_{2,x}$ in Eq. (21)

into linear in the small parameter $a_2/\omega_0$ expressions, similar to high field approximation in conventional magnetic resonance, and convert Eq. (21) to a much simpler expression

$$R_{1,m} + R_{1,m-1} = R_{2,n} + R_{2,n-1} = \omega_0 + na_2. \tag{22}$$

This is equivalent to a cubic equation, which is linearized further by introducing a second small parameter $a_2/a_1$ to obtain the sought solution:

$$\omega_0^* = -a_1 \frac{I_1(I_1+1)}{2m} - a_2 n \left( \frac{I_1(I_1+1)}{2m^2} - \frac{1}{2I_1(I_1+1)} \right). \tag{23}$$

This is valid for each pair of nuclear spin projections $(m,n)$, but since we consider the crossings in positive fields, as in Eq. (7), we should formally restrict $m$ to be in the range $-I < m < 0$, while $n$ can assume any of its $2I_2+1$ possible values.

Tracing the two-step linearizing high field assumption for the second partner back to the starting expression of Eq. (18), it is readily seen that if the second partner contains an arbitrary set of magnetic nuclei with HFC so small that the 145 high field limit is valid at fields of Eq. (7) for its entire ESR spectrum in conventional sense, we can set from the beginning

$$p_2(t) = 1, \quad h_2(t) = \exp\left[ i \left( \omega_0 + \sum_{k,n_k} a_k n_k \right) t \right], \tag{24}$$

which in the same order produces

$$R_{1,m} + R_{1,m-1} = R_{2,n} + R_{2,n-1} = \omega_0 + \sum_{k,n_k} a_k n_k, \tag{25}$$

where $a_k$ and $n_k$ are the HFC constants and spin projections for $k$-th nucleus. By the same token, an inhomogeneous 150 spectrum, like a "semiclassical" Gaussian shape (Schulten and Wolynes, 1978), can be used in place of the sum $\sum_{k,n_k} a_k n_k$. Substituting Eq. (25) into Eq. (21) as a result of the first step of linearization, we obtain Eq. (22) with the term $na_2$ changed for the sum $\sum_{k,n_k} a_k n_k$. Solving it by the second step of linearization, we arrive at the result similar to Eq. (23):

$$\omega_0^* = -a_1 \frac{I_1(I_1+1)}{2m} - \left( \frac{I_1(I_1+1)}{2m^2} - \frac{1}{2I_1(I_1+1)} \right) \sum_{k,n_k} a_k n_k. \tag{26}$$

This is the central result of this work, and its interpretation is as follows: provided the entire ESR spectrum of the second 155 partner is compact enough in comparison to the hyperfine coupling in the dominant first partner, each characteristic level-crossing "line at multiple field" of Eq. (7) spells out the ESR spectrum of the second partner, scaled in field by a constant factor, which depends on the specific crossing and is given in parentheses in Eq. (26), with intensity of Eq. (16) borrowed from the original crossing and distributed over the spectrum as in the conventional ESR. We also note that the field scaling factor in Eq. (26) is identical to the scaling factor for the homogeneous width in Eq. (17), as both are ultimately determined 160 by the relative slopes of the linearized crossing levels, so the scaling is uniform from both homogeneous and inhomogeneous perspective. The sum $\sum_{k,n_k} a_k n_k$ can be substituted for any spectral shape function $F(\omega_0)$, provided that it is restricted to linear, first order spectrum in terms of conventional ESR. Second-order conventional ESR spectra (Fessenden, 1962) would not carry transparently through the double step linearization procedure and would have required a more careful treatment to second order at both steps.

We finally note that the same formalism can be used to analyze the level crossings driven by substantial difference in $g$-values of the pair partners together with HFC, mentioned in the introduction and studied in detail in (Brocklehurst, 1999). Assuming that the first partner has one spin-$I$ nucleus with HFC $a_1$ and $g$-value $g_1$, while the second partner has no magnetic nuclei, but a shifted $g$-value $g_2$, and introducing relative shift of $g$-values $\delta = \dfrac{g_2 - g_1}{g_1} \ll 1$, we should set for the second partner

$$p_2(t) = 1, \quad h_2(t) = \exp\left[i\omega_0(1+\delta)t\right], \tag{27}$$

yielding

$$R_{2,n} + R_{2,n-1} = \omega_0(1+\delta) \tag{28}$$

and two sets of solutions:

$$\omega_{01}^* = -a_1 \frac{I_1(I_1+1)}{2m}, \quad \omega_{02}^* = a_1 \frac{I_1(I_1+1)}{2m} + a_1 \frac{m}{\delta}. \tag{29}$$

While the first set coincides with Eq. (7), or Eq. (26) with $a_2$ set to zero, and the lines at $\omega_{01}^*$ can be understood as lines in weak fields where the difference in $g$-values is yet not consequential, the second set has small parameter $\delta$ in denominator and gives the same lines translated to high fields. Expressions of Eq. (29) were first derived in (Brocklehurst, 1999) and are rederived here only to show the equivalence of the employed approach, and the reader is referred to (Brocklehurst, 1999) for a more in-depth discussion of $\Delta g$-induced level crossings.

## 3 Even number of equivalent spin-$\frac{1}{2}$ nuclei to drive spin evolution in the pair

Several comments regarding the results of the previous section are now in order. First of all, the "driving" crossings of Eq. (7) require a nucleus with spin $I > 1$ and substantial HFC, that would furthermore not compromise the relaxation properties of the recombining pair. Although nuclei with spins $\frac{3}{2}$ and higher, like $^{35,37}$Cl ($\frac{3}{2}$) (Bagryansky et al., 1998), $^{27}$Al ($\frac{5}{2}$), $^{69,71}$Ga($\frac{3}{2}$), $^{113,115}$In ($\frac{9}{2}$) (Sergey et al., 2012), $^{73}$Ge($\frac{9}{2}$) (Shokhirev et al., 1991; Borovkov et al., 2003) occasionally occur in magnetic field effect experiments, so far the only resolved lines in multiple fields of Eq. (7) have been reported for systems containing sets of equivalent spin-$\frac{1}{2}$ nuclei, either protons or fluorines (Stass et al., 1995b; Saik et al., 1995; Grigoryants et al., 1998; Kalneus et al., 2006a). The best results making them promising for such applications were obtained for radical anions of either hexafluorobenzene (six fluorines with $a$=13.7 mT) or octafluorocyclobutane (eight fluorines with $a$=15.1 mT) paired with a narrow partner radical cation. This means that the single spin-$I$ nucleus would in most cases be an

190 effective spin equal to one of the possible values of the total spin for a set of equivalent spin-$\frac{1}{2}$ nuclei, with the corresponding statistical distribution, and thus there would be a corresponding composite level-crossing spectrum with contributions from all possible values of the total nuclear spin.

It's a fluke that in the most common case of an even number of spin-$\frac{1}{2}$ nuclei the crossings of Eq. (7) occur at simple integer multiples of the HFC constant $a$ and mostly overlap to reinforce each other, but the downside is that the 195 overlapping spectra have different field scaling factors. However, in reality the latter does not create that much of a problem. Let's take hexafluorobenzene with its six equivalent fluorines as a typical example. The possible values of pairs $(I, |m|)$ to produce crossings of Eq. (7) would be $(3, 2)$, $(3, 1)$, and $(2, 1)$, producing the lines of Eq. (7) in the fields $3a$, $6a$, and $3a$, respectively. The corresponding field scaling factors from Eq. (26) would be $\frac{12}{8} - \frac{1}{24}$, $\frac{12}{2} - \frac{1}{24}$, and $\frac{6}{2} - \frac{1}{12}$, respectively. It can be seen that the two overlapping lines at $3a$ have different scaling factors, reflecting the different slopes of the 200 intersecting energy levels. Now let's estimate their relative contributions. The statistical weights of subensembles with total spin $I$ for a set of an even number $n$ of spin-$\frac{1}{2}$ nuclei $W(I; n)$ can be taken from (Bagryansky et al., 2000):

$$W(I; n) = \frac{(2I+1)^2 n!}{2^n \left(\frac{n}{2} - I\right)! \left(\frac{n}{2} + I + 1\right)!}, \tag{29}$$

and in our example evaluate to $W(3; 6) = \frac{7}{64}$ and $W(2; 6) = \frac{25}{64}$. It can be seen that the overlapping crossing at $3a$ is statistically dominated by the smaller total spin $I = 2$, while the higher total spin $I = 3$ is responsible for the crossing at 205 $6a$. Omitting the small corrections of $\frac{1}{24}$ and $\frac{1}{12}$, the field scaling factors for the crossings at $3a$ and $6a$ are 3 and 6, respectively, with the crossing at $3a$ being nearly 4 times stronger and twice narrower, which is critical in field modulation experiments.

For our second example of 8 equivalent fluorines we would get the weights of $W(4; 8) = \frac{9}{256}$, $W(3; 8) = \frac{49}{256}$, and $W(2; 8) = \frac{100}{256}$, and lines at $10a$, $5a$, and $10/3a$ from $I = 4$ in addition to already described lines at $3a$ and $6a$. Again the 210 strongest line at $3a$ is dominated by the contribution from the $I = 2$ subensemble with field scaling factor 3 and swamps the much weaker nearby line at $10/3a$ coming from $I = 4$, the line at $6a$ is dominated by $I = 3$ with the scaling factor of 6 and swamps the nearby line at $5a$ from $I = 4$, and the only genuinely new line from $I = 4$ is the line at $10a$ with the scaling factor of 10. To generalize this, we note that the "dominant" lines come from pairs $(I, |m|)$ with all possible values of $I$ in

the range $1 < I \le n/2$ and $|m| = 1$. Comparing the expressions for the positions $\omega_0^*$ of the crossing peaks and the field scaling factors $f$ in Eqs. (26,17) and omitting the small correction $\left(2I_1\left(I_1+1\right)\right)^{-1}$ in the scaling factors, we see that

$$\text{for } |m| = 1 \quad \left\{\frac{\omega_0^*}{a_1} = \frac{I_1\left(I_1+1\right)}{2|m|}\right\} = \left\{f = \frac{I_1\left(I_1+1\right)}{2m^2}\right\} = \frac{I_1\left(I_1+1\right)}{2}. \tag{30}$$

From this we derive our Simple Rule of Structure:

*Given a radical pair with $n = 2k$ equivalent spin-$\frac{1}{2}$ nuclei with a large HFC constant a in one partner to drive spin evolution and a compact relative to HFC constant a ESR spectrum in the other partner, expect in the magnetic field effect curve $k-1$ progressively weaker copies of the ESR spectrum of the narrow partner at fields $\omega_0^* = f_q a$ scaled in field by $f_q$, $f_q = q\left(q+1\right)/2$, $q = 2,\ldots k$. The strongest copy is the lowest of them, for f=3, i.e., it is at "triple field" and is "triply scaled".*

Although so far the only experimental observations of the ESR structure using this approach have been the "spectrum" at $3a$ for an unresolved inhomogeneous spectrum as proof of principle in (Tadjikov et al., 1996) and arguments based on the lack of the inhomogeneous spectrum at $3a$ in several works on radiation chemistry (Tadjikov et al., 1997; Usov et al., 1997; Sviridenko et al., 1998) from just one group, we hope that the current surge of interest to the level (anti)crossing interpretation of magnetic resonance will draw attention to this aspect of the humble magnetic field effect experiments.

**4 Other possible configurations of the driving spins**

Although the case of an even number of driving spins-$\frac{1}{2}$ is the most convenient, it is not the only possible one. Still staying with equivalent nuclei, one experimental case of three spins-$\frac{1}{2}$ has been reported, for the radical anion of 1,3,5-trifluorobenzene complemented with a partner with a narrow ESR spectrum (Kalneus et al., 2006a). The only level-crossing line here comes from the effective total spin $I = \frac{3}{2}$ for three fluorines, as expected, and could in principle be used as a vehicle to obtain the ESR spectrum of the partner. Furthermore, tracing back how the structure-bearing partner was introduced after linearization in Eqs. (24,25), we see that there is nothing special in the single spin-$I$ or equivalent spin-$\frac{1}{2}$ nuclei other than the possibility to treat them analytically and obtain a well-defined level-crossing line if the HFC coupling is sufficiently strong. Of course, this is a rather substantial "other than", but it does not exclude other possible spin systems as the driving partner if they appear.

And such systems do indeed exist. Several experimental reports of the resolved MARY spectra for systems with not equivalent nuclei with large HFC constants, in all cases fluorines, have been published. These include radical anions of 1,2,3-trifluorobenzene (Kalneus et al., 2007), pentafluorobenzene (Kalneus et al., 2006b), and recently several fluorosubstituted diphenylacetylenes (Sannikova et al. 2019), again complemented with a radical cation having a narrow ESR spectrum. The spectra featured well-defined lines that were reproduced in simulations and were traced to clusters of

level crossings in the spin system of the pair. Although the "multiplication of ESR spectrum" of the possible pair partner would in this case be not very informative due to high concentration of close and overlapping level crossings, it would be rather important to at least keep in mind the inhomogeneous broadening of these lines due to hyperfine couplings in the second partner.

     Analysis of the level-crossing spectra for systems with non-equivalent nuclei also helped develop the concept of

"active crossings" (Pichugina and Stass, 2010) as a substitute for traditional selection rules for transitions in conventional magnetic resonance. To put it simply, of all the energy level crossings present in the spin system of the pair only those between levels reachable from the same initial (singlet) state of the pair may produce observable lines due to interference of coherently populated eigenstates. In terms of the discussion of this work the active crossings would be the crossings of levels from the same 4-dimensional blocks with energies of Eq. (3), to which correspond the terms with fixed $m$ in the sums of

Eq. (13).

## 5 Introducing nuclei into the driving partner: crossings *vs.* anticrossings

The transparency of translating the ESR spectrum of the narrow partner to the level-crossing line due to the partner with strong HFC is rather amazing, and is a consequence of separating these two roles and adding the new nuclei to the partner that originally just complements the pair. This can be more easily understood using the language of wave functions rather

than density matrix as follows. Suppose we have an active level crossing of Eq. (7) from a subspace of pair eigenstates of Eq. (3) spanning four functions of the product basis $\left| S_{1z}, I_z \right\rangle_1 \left| S_{2z} \right\rangle_2$ with projections $\left| \alpha, m \right\rangle_1 \left| \beta \right\rangle_2$, $\left| \alpha, m-1 \right\rangle_1 \left| \alpha \right\rangle_2$, $\left| \beta, m \right\rangle_1 \left| \alpha \right\rangle_2$, $\left| \beta, m+1 \right\rangle_1 \left| \beta \right\rangle_2$. The only non-secular interaction in the pair is hyperfine coupling in the first partner, which means that the eigenstates of the pair will be of the form $\left| \xi_i \right\rangle_1 \left| S_{2z} \right\rangle_2$, still remaining the products of functions for the two partners. The energies of the eigenstates will be the sums of energies for the two partners, and the nontrivial spin evolution

leading to level crossing lines is due to simultaneously projecting the starting singlet state onto several eigenstates at the moment of pair creation and back at the moment of recombination, and beating due to different energies of the populated eigenstates that partially stops when some energies become equal, *i.e.*, some levels cross.

     Now let us introduce nuclei to the second partner, *i.e.*, augment its eigenstate $\left| S_{2z} \right\rangle_2$ to include the indices of the newly introduced nuclear spin projections to $\left| S_{2z}, n_{1z}, \ldots, n_{kz} \right\rangle_2$. Since we are in the conditions of the high field limit for the

second partner, as in conventional ESR, the augmented eigenfunctions will in fact be products of electron and nuclear functions $\left|S_{2z}\right\rangle_2 \left|n_{1z}, \ldots, n_{kz}\right\rangle$, splitting in energy by the corresponding secular contribution $\pm \frac{1}{2} \sum_{k,n_k} a_k n_k$. The states of the newly introduced nuclei in this approximation are not affected by spin evolution in the pair, and thus the nuclear function $\left|n_{1z}, \ldots, n_{kz}\right\rangle$ effectively becomes a new conserved multi-index, by which the state space for the pair augmented with new nuclei is partitioned. The original 4-dimensional subspaces housing the active crossings are multiplied into copies differing

only by the new multi-index, each giving the same active crossing, but at a correspondingly shifted energy and with a proportionally reduced intensity borrowed from the original crossing. The varying scaling with the field comes from the different relative slopes of the linearized crossing levels, which now differ from the $\pm \omega_0/2$ of conventional ESR and become progressively more shallow with increasing external field, spreading the same vertical shift in energy to a progressively wider horizontal scaling with the field. The same can be said about the scaling of the homogeneous

contribution to linewidth of Eq. (17), which is converting the same width of the energy levels due to finite lifetime into the width along the field axis. Since this multiplication of state subspaces is entirely due to the second partner, this discussion applies to any hyperfine structure of the driving partner, provided its HFC constants are sufficiently high.

The situation with adding the new nuclei to the first, driving, partner is quite different. Now the function augmented with additional nuclear spins is not of the high field limit case, and effectively a new interaction is added into a coupled spin

system. Let's again turn to the wavefunction illustration, first for single nuclear spin-$\frac{1}{2}$ and just one added spin-$\frac{1}{2}$ nucleus with a small HFC constant $a_2 \ll a_1$. The original functions $\left|S_{1z}, I_z\right\rangle_1$ for the first partner are now augmented to functions $\left|S_{1z}, I_z, n\right\rangle_1$, which do not factor into simple product, and the newly introduced index $n$ is not just an external conserved quantity. Instead we have introduction of (weak) additional interactions into a system of crossing levels, which leads to anticrossings. Since the total spin projection is conserved for each radical, e.g., functions $\left|\alpha, \alpha, \beta\right\rangle_1$, $\left|\alpha, \beta, \alpha\right\rangle_1$,

$\left|\beta, \alpha, \alpha\right\rangle_1$ now fall into one sub-block of the Hamiltonian and are mixed together, and we note that without the added nucleus the first of them and the two other were in different blocks, and would have contributed to different active crossings. Now addition of a weak new coupling introduces an anticrossing, possibly between different blocks, instead. The key questions are now what anticrossings are being introduced, and whether the original crossings turn into anticrossings upon addition of the new interaction. This situation must be familiar to experts in hyperpolarized NMR in the form of level

anticrossings in three-spin systems, where one nucleus is J-coupled to two other nuclei (Miesel et al., 2006; Pravdivtsev et al. 2013). Another close example is a three-spin system biradical-ion/radical ion with exchange interaction within the biradical and hyperfine interaction with a nucleus in either partner (Lukzen et al., 2002; Verkhovlyuk et al. 2007), where a nucleus in the biradical ion produces an anticrossing near the main line of J-resonance in the biradical, while a nucleus in the

radical partner produces a crossing. Similar dichotomy is also observed in magnetic effects in a biradical/stable radical complex with different distributions of inter- and intra-partner exchange interactions (Magin et al. 2004; 2005; 2009).

To analyze the resulting changes in eigenstructure let us review Eq. (3). The expressions for energies are clearly of the form $\left(-\dfrac{a}{4} \pm R_m\right) \pm \dfrac{\omega_0}{2}$ and are the sums of the energies of two independent partners. One of them has coupled electron and nuclear spins and corresponds to the first term, which is the familiar Breit-Rabi expression (Breit and Rabi, 1931) for arbitrary nuclear spin $I$. The other partner has just electron spin. We also require that the states of Eq. (3) be reachable from the same electron spin singlet state. To obtain the pair state subspace with total spin projection $\Sigma_z = m$ we thus need to combine two states of the first partner with total projection $M_z = m + \frac{1}{2}$ spanning the product basis states $|\alpha, m\rangle_1, |\beta, m+1\rangle_1$ with the $|\beta\rangle_2$ state of the second partner, and two states of the first partner with total projection $M_z = m - \frac{1}{2}$ spanning the product basis states $|\alpha, m-1\rangle_1, |\beta, m\rangle_1$ with the $|\alpha\rangle_2$ state of the second partner. So the energies of Eq. (3) correspond to the following functions:

$$
\begin{aligned}
&E_1(m) = \left(-\frac{a}{4} + R_m\right) - \frac{\omega_0}{2}, \quad |\psi_1(m)\rangle = \left(\cos_m |\alpha, m\rangle_1 + \sin_m |\beta, m+1\rangle_1\right)|\beta\rangle_2, \\
&E_2(m) = \left(-\frac{a}{4} - R_m\right) - \frac{\omega_0}{2}, \quad |\psi_2(m)\rangle = \left(-\sin_m |\alpha, m\rangle_1 + \cos_m |\beta, m+1\rangle_1\right)|\beta\rangle_2, \\
&E_3(m) = \left(-\frac{a}{4} + R_{m-1}\right) + \frac{\omega_0}{2}, \quad |\psi_3(m)\rangle = \left(\cos_{m-1} |\alpha, m-1\rangle_1 + \sin_{m-1} |\beta, m\rangle_1\right)|\alpha\rangle_2, \\
&E_4(m) = \left(-\frac{a}{4} - R_{m-1}\right) + \frac{\omega_0}{2}, \quad |\psi_4(m)\rangle = \left(-\sin_{m-1} |\alpha, m-1\rangle_1 + \cos_{m-1} |\beta, m\rangle_1\right)|\alpha\rangle_2,
\end{aligned}
\tag{31}
$$

where trig notation was adopted for the mixing coefficients in the Breit-Rabi functions.

Now let us introduce an additional nucleus with spin $K$ with a weak hyperfine coupling into the first partner by building product functions of the form $|\psi_i(m)\rangle|n\rangle$ and treating the new hyperfine interaction as perturbation $\hat{V} = b\vec{S}_1\vec{K}$. We recall that the original active crossings were the ones within the blocks of Eq. (31) for $E_1 = E_3, E_2 = E_4$ in zero field and $E_1 = E_4$ in the fields of Eq. (7). Now we note that the perturbation is diagonal with respect to second electron spin and thus has zero matrix elements $V_{13}, V_{24}, V_{14}$ between the required functions, and conclude that the original active crossings all survive and do not turn into anticrossings.

Not vanishing matrix elements can be obtained between function of adjacent blocks of Eq. (31), e.g., $|\psi_{1,2}(m)\rangle$ and $|\psi_{1,2}(m-1)\rangle$:

$$\langle \psi_1(m);n|\hat{V}|\psi_1(m-1);n+1\rangle = \cos_m \sin_{m-1} \langle \alpha,m;n|\frac{b}{2}S_{1+}K_-|\beta,m;n+1\rangle_1$$

(32)

$$= \cos_m \sin_{m-1} \frac{b}{2}\sqrt{(K+n+1)(K-n)},$$

with similar results for all four combinations of indices 1,2 for the two functions, and all four combinations for the indices 3,4 of the other two functions. This implies anticrossings if the original functions corresponded to crossing energy levels. Looking at expressions for energies of Eq. (31), this in turn implies $R_m = \pm R_{m-1}$, which is indeed possible in zero field in the variant $R_m = R_{m-1}$. Thus, for each pair of adjacent 4-dimensional blocks of Eq. (31) with $\Sigma_z = m$ and $\Sigma_z = m-1$ we

had four crossings in zero field for $E_i(m) = E_i(m-1), i = 1,\ldots,4,$ that turn into the respective anticrossings. Furthermore, these anticrossings stitch together all the subspaces of the initially partitioned state space. Note that the original crossings were not active, they corresponded to different subspaces and thus their states would not be populated simultaneously and interfere. However, as opposed to the interference effects of genuine level crossings, the anticrossings just reshape the energy level layout and do not require the simultaneous population of the contributing states, and thus the added weak

hyperfine interaction in the first partner turns dormant crossings into acting anticrossings in zero field. We further note that the specific hyperfine structure for the added nuclei is not important, it just suffices that they provide the nonzero couplings of Eq. (32) between the zero-field states of the adjacent blocks.

Since the newly introduced weak interaction does not affect the original active crossings, we may evaluate its effect on energy levels to first order by evaluating the average values of the perturbing interaction for product functions

$|\psi_i(m)\rangle|n\rangle$, and for the interesting case of the crossing $E_1 = E_4$ we obtain

$$V_{11} = b\frac{n}{2}\left(2\cos_m^2 - 1\right), V_{44} = -b\frac{n}{2}\left(2\cos_{m-1}^2 - 1\right).$$
(33)

This moves the surviving active crossing in energy by

$$\Delta E_{1,4} = V_{11} - V_{44} = bn\left(\cos_m^2 + \cos_{m-1}^2 - 1\right)$$
(34)

evaluated at the crossing field of Eq. (7), which is then converted to shift in field by the field scaling factor due to differential

slope taken from Eq. (26). Such evaluation yields some unwieldy expression hardly qualifying for a "simple rule" and of little practical utility, but note that the shift in field is simply proportional to $bn$ with the scaling factor depending only on the properties of the "driving" spins and particular crossing, and again this linearity means that the spectrum corresponding to the added weaker hyperfine structure will be spelled out at the original crossing point of Eq. (7), as was the case for Eq. (26). In practice this means that additional nuclei with smaller HFC constants in the driving partner at least contribute an

inhomogeneous broadening to the level crossing line complicating its experimental observation, as is the case for radical anion of 1,2,4,5-tetrafluorobenzene containing two protons with smaller couplings in addition to four equivalent fluorines

(Kalneus et al., 2006a). The reason for the noted linearity is of course the applicability of first order perturbation theory due to surviving of the active crossings. Such simple considerations can sometime help advance in a problem that seem otherwise overwhelming (Stass, 2019).

Similar issues of "localization of interaction" in pair partners also arise in the discussion of $\Delta g$ -induced resonances (Brocklehurst, 1999) and in the discussion of the zero field line in magnetic field effect curve, where it was mentioned several times that distribution of HFC over both partners as opposed to their concentration in one partner decreases the magnitude of the effect (Timmel et al., 1998; Kalneus et al., 2005; Woodward et al., 2008). In our picture it appears as arising of anticrossings at zero field that spread and counteract the active crossings originally present there, additionally
washing away the well-defined partitioning into state subspaces with pronounced state interference. Another place where the crossing *vs.* anticrossing discussion is very relevant is the so-called *J*-resonance in radical pairs (Hamilton et al., 1989; Shkrob et al., 1991) or linked donor-acceptor dyads (Weller et al., 1984; Ito et al., 2003; Wakasa et al., 2015; Steiner et al., 2018), where exchange coupling between the two partners shifts the triplet electron spin manifold relative to singlet, and at certain magnetic field the singlet term crosses with one of the triplet sublevels. In many cases these crossing turns into an
anticrossing due to additional weaker interactions, such as HFC with magnetic nuclei, but traditionally the situation is often still referred to as "ST.-crossing", even though the technical discussion clearly identifies it as anticrossing.

From the practical viewpoint the important difference between crossings and anticrossings is that the former partially block spin evolution due to state interference, and thus lock the pair in its initial state, while the latter accelerate spin evolution and assist in leaving the initial state. Furthermore, using the settings of this work as an example, while the
crossings produce sharp lines with widths of the order of inverse lifetime $\tau^{-1}$ separated by intervals of the order of introduced interaction $V$, the anticrossings produce much broader lines of the opposite phase with widths of the order of $\sqrt{\tau^{-2}+V^2}$. If we introduce three parities, to indicate the initial state $\Gamma_i = +1$ for singlet and $-1$ for triplet, the observation state $\Gamma_o = +1$ for singlet and $-1$ for triplet, and the type of crossing $\Gamma_c = +1$ for crossing and $-1$ for anticrossing, then we can derive our Even Simpler Rule of Signs:


*The sign of a feature in a level crossing spectrum is given by* $\Gamma = \Gamma_i \bullet \Gamma_o \bullet \Gamma_c$.

## 6 Level crossing lines as flip-flop resonance in pair partners

Creation of a spin-correlated radical (ion) pair is a shock excitation for a radical pair Hamiltonian, and, as any shock-excited quantum system, the pair "rings" at its eigenfrequencies (Salikhov, 1993). Since for the pairs of this work the Hamiltonian of
the pair is a sum of independent Hamiltonians for the two partners, the ring frequencies must be some linear combinations of the eigenfrequencies for the pair partners. It is clear that creating the pair in singlet state with a given nuclear configuration

must select some subset of the possible ring frequencies, and the examples discussed in this work provide some very useful insight regarding this selection.

Let us again review the expressions of Eq. (31) for energies/functions of the typical subspace of radical pair spin system. The condition $E_1 = E_4$ for the level crossing line can be trivially rearranged as

$$\left(-\frac{a}{4} + R_m\right) - \left(-\frac{a}{4} - R_{m-1}\right) = \omega_0, \tag{35}$$

where at the LHS we have the difference of energies of two eigenstates of the first pair partner, and at the RHS an equivalent difference for the second partner. Now we look at the corresponding functions $\psi_{1,4}$ and recall that our pair starts from and recombines to a singlet state with nuclear spin projection $m$, which is function $\left(|\alpha, m\rangle_1 |\beta\rangle_2 - |\beta, m\rangle_1 |\alpha\rangle_2\right)/\sqrt{2}$. We note the following correlation between "transitions" between functions $\psi_{1,4}$ and changes in the spin states of the individual radicals:

$$\psi_1 \leftrightarrow \psi_1 \quad \text{means} \quad |\alpha, m\rangle_1 \leftrightarrow |\beta, m\rangle_1 \quad and \quad |\beta\rangle_2 \leftrightarrow |\alpha\rangle_2. \tag{36}$$

The two latter relations mean an allowed ESR transition in the first radical and simultaneously an opposing allowed ESR transition in the second radical, at the same frequency given by the differences in the energies of the corresponding true eigenstates of each of the pair partners. The statement about "allowed ESR transition" should be understood as transition induced by nonzero matrix element of electron spin operator, $e.g.$, $S_{1x}$ (for the first partner), between the factors of the functions pertaining to this partner, and for functions $\psi_{1,4}$ this is reduced to nonzero matrix element between the functions of Eq. (36). Therefore in this case the level crossing line appears in the field where such a flip-flop energy conserving transition in the pair partners can occur. Inspection of functions in Eq. (31) demonstrates that the same also turns out to be true for the two level crossings in zero field that correspond to conditions $E_1 = E_3$ and $E_2 = E_4$.

Now let us consider the case of compact ESR structure at the second partner, for which the level crossing condition of Eq. (25) can again be slightly rearranged to give

$$\left(-\frac{a}{4} + R_m\right) - \left(-\frac{a}{4} - R_{m-1}\right) = \omega_0 + \sum_{k,n_k} a_k n_k. \tag{37}$$

The functions for the subspace with the conserving nuclear configuration of the narrow partner are now given by expressions of Eq. (31) with all functions multiplied by the conserved multi-index $|n_{1z}, \ldots, n_{kz}\rangle$. The previous paragraph can be repeated nearly word for word with the conclusion that the level crossing lines appear in the field where simultaneous flip-flop energy conserving allowed ESR transitions in the pair partners can occur, between the entangled electron-nuclear energy levels of the first partner and between the conventional high-field limit decoupled energy levels in the second radical.

Finally, let us consider the case of both partners containing a single nucleus with arbitrary spin without any assumptions on the relative sizes of their HFC constants $a_{1,2}$. The level crossing condition for this case is given by Eq. (21), which can again be rearranged into

$$\left(-\frac{a_1}{4}+R_{1,m}\right)-\left(-\frac{a_1}{4}-R_{1,m-1}\right)=\left(-\frac{a_2}{4}+R_{2,n}\right)-\left(-\frac{a_2}{4}-R_{2,n-1}\right), \tag{38}$$

expressing the equality of transition frequencies for the two partners. To build the functions for a subensemble with $\Sigma_z = m+n$ reachable from the same singlet state $\left(\left|\alpha,m\right\rangle_1\left|\beta,n\right\rangle_2-\left|\beta,m\right\rangle_1\left|\alpha,n\right\rangle_2\right)/\sqrt{2}$ similar to Eq. (31), we need to combine the Breit-Rabi functions for the first partner with total projection $M_z = m+\frac{1}{2}$ with the Breit-Rabi functions for the second partner with total projection $N_z = n-\frac{1}{2}$ and, vice versa, functions with $M_z = m-\frac{1}{2}$ with functions with $N_z = n+\frac{1}{2}$, to obtain function sets

$$\begin{cases}\cos_m\left|\alpha,m\right\rangle_1+\sin_m\left|\beta,m+1\right\rangle_1\\-\sin_m\left|\alpha,m\right\rangle_1+\cos_m\left|\beta,m+1\right\rangle_1\end{cases}\times\begin{cases}\cos_{n-1}\left|\alpha,n-1\right\rangle_2+\sin_{n-1}\left|\beta,n\right\rangle_2\\-\sin_{n-1}\left|\alpha,n-1\right\rangle_2+\cos_{n-1}\left|\beta,n\right\rangle_2\end{cases},$$

$$\begin{cases}\cos_{m-1}\left|\alpha,m-1\right\rangle_1+\sin_{m-1}\left|\beta,m\right\rangle_1\\-\sin_{m-1}\left|\alpha,m-1\right\rangle_1+\cos_{m-1}\left|\beta,m\right\rangle_1\end{cases}\times\begin{cases}\cos_n\left|\alpha,n\right\rangle_2+\sin_n\left|\beta,n+1\right\rangle_2\\-\sin_n\left|\alpha,n\right\rangle_2+\cos_n\left|\beta,n+1\right\rangle_2\end{cases}. \tag{39}$$

The energy matching condition of Eq. (38) corresponds to the following functions:

$$\left(\cos_m\left|\alpha,m\right\rangle_1+\sin_m\left|\beta,m+1\right\rangle_1\right)\times\left(-\sin_{n-1}\left|\alpha,n-1\right\rangle_2+\cos_{n-1}\left|\beta,n\right\rangle_2\right),$$

$$\left(-\sin_{m-1}\left|\alpha,m-1\right\rangle_1+\cos_{m-1}\left|\beta,m\right\rangle_1\right)\times\left(\cos_n\left|\alpha,n\right\rangle_2+\sin_n\left|\beta,n+1\right\rangle_2\right). \tag{40}$$

We see that again the level crossing line corresponds to a simultaneous energy-conserving flip-flop transition

$$\left|\alpha,m\right\rangle_1\leftrightarrow\left|\beta,m\right\rangle_1\quad\text{and}\quad\left|\beta,n\right\rangle_2\leftrightarrow\left|\alpha,n\right\rangle_2 \tag{41}$$

in the two pair partners that correspond to allowed ESR transitions in the opposite directions.

We should not try to generalize these observations beyond what can be established from results derived in this work, but the pattern is quite obvious, and therefore we suggest for further consideration and discussion a Provisional Rule of Resonances:

*The level crossing lines appear in the fields where simultaneous energy-conserving ESR allowed flip-flop transitions can proceed in the pair partners.*

The idea that simultaneous transitions in spin systems of pair partners can lead to level-crossing lines probably goes back to work (Brocklehurst, 1999), and a similar result was also obtained for interference of ESR transitions in the ESR (RYDMR) spectra of radical pairs in (Salikhov et al., 1997; Tadjikov et al., 1998).


## 7 Compendium of typical resolved spectra

In this section we present several figures illustrating typical resolved level-crossing spectra that could be reasonably expected in experiment. For all figures the driving partner with large HFC constants mimics hexafluorobenzene radical anion and has six equivalent spin-$\frac{1}{2}$ nuclei with HFC constant $A$, while the second partner contains two equivalent or

nonequivalent spin-$\frac{1}{2}$ nuclei, as indicated. All parameters, i.e., the external magnetic field, the smaller couplings in the second partner, and the recombination parameter $s$, are measured in the units of $A$.

Figure 1 shows a review spectrum for a pair with equivalent nuclei in both partners that can be calculated analytically in the full field range from zero to well past the level-crossing lines. In this case the smaller couplings are taken as one-tenth of the large ones, and the spectrum fully conforms to expectations as discussed in this work.

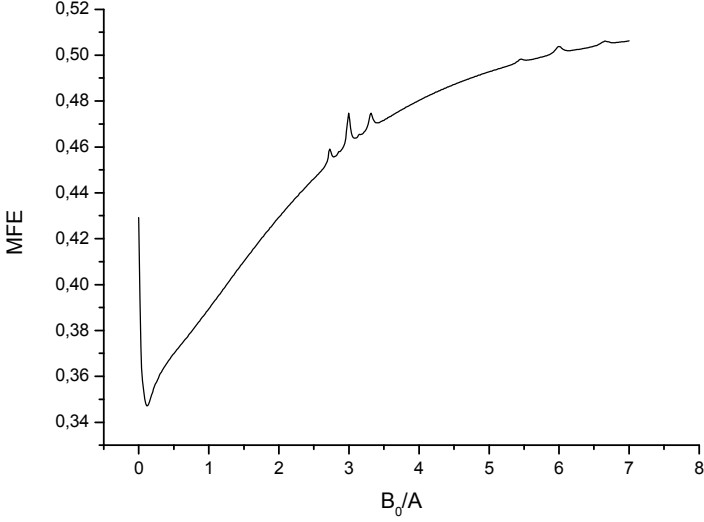


**Figure 1: The review MFE curve for a pair with 6 equivalent spin-$\frac{1}{2}$ nuclei with HFC constant $A$ in one partner and 2 equivalent spin-$\frac{1}{2}$ nuclei with HFC constant $A/10$ in the second partner, recombination parameter $s = A/100$.**

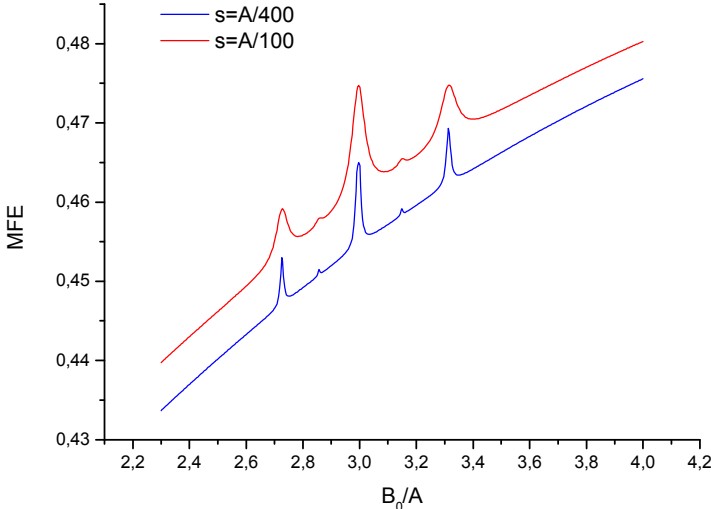

**Figure 2: Closeup of the spectrum from Fig. 1 in the vicinity of $B_0 = 3A$ for two values of the recombination parameter**
*s = A/100, s = A/400.*

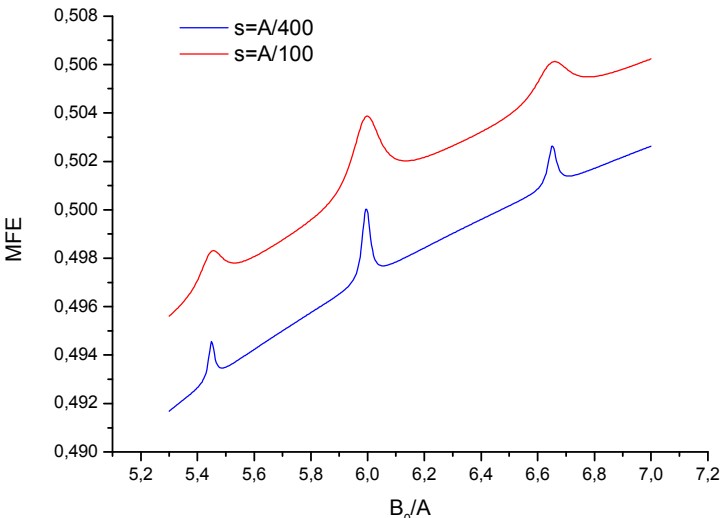

**Figure 3: Closeup of the spectrum from Fig. 1 in the vicinity of $B_0 = 6A$ for two values of the recombination parameter**
*s = A/100, s = A/400.*

450        Figures 2 and 3 show in more details the regions of the level crossing lines at $3A$ and $6A$ for the parameters used in Fig. 1, as well as for a 4-fold reduced recombination parameter, i.e., for a longer lived pair, to increase resolution. Note the familiar 1-2-1 pattern for two equivalent spin-$\frac{1}{2}$ nuclei with splittings equal to $3A/10$ and $6A/10$, as expected. Also note a pair of lower-intensity lines with half the splitting in Fig. 2, corresponding to the minor contribution of the subensemble with total nuclei spin of the "driving" partner $I = 3$ with the scaling factor 12/8 instead of 3 to the level crossing line at triple

HFC constant.

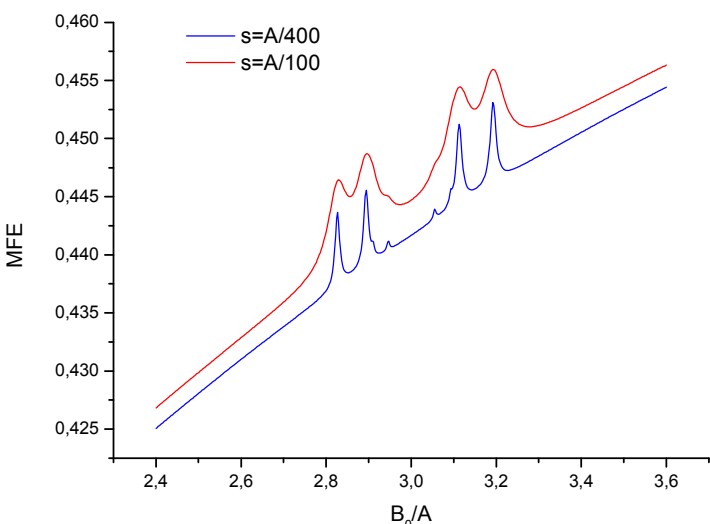

**Figure 4: Region in the vicinity of** $B_0 = 3A$ **for a pair with 6 equivalent spin-$\frac{1}{2}$ nuclei with HFC constant $A$ in one partner and 2 non-equivalent spin-$\frac{1}{2}$ nuclei with HFC constant $A/10$ and $A/40$ in the second partner, recombination parameter $s = A/100$,**

$s = A/400$**.**

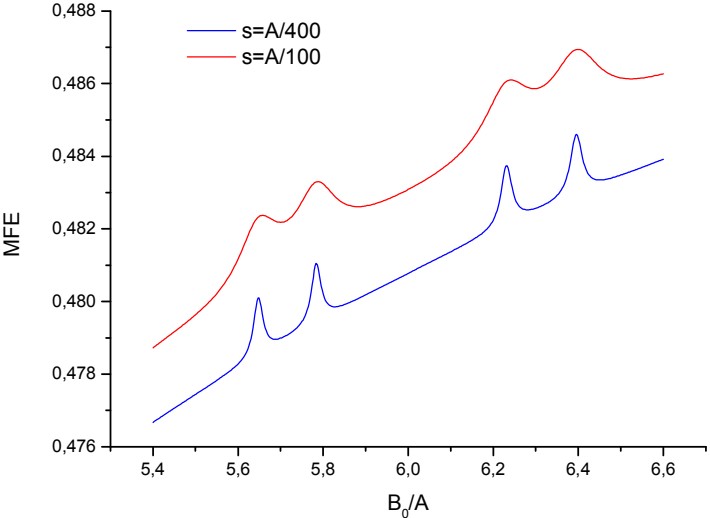

**Figure 5: Region in the vicinity of $B_0 = 6A$ for a pair with 6 equivalent spin-$\frac{1}{2}$ nuclei with HFC constant $A$ in one partner and 2 non-equivalent spin-$\frac{1}{2}$ nuclei with HFC constant $A/10$ and $A/40$ in the second partner, recombination parameter $s = A/100$, $s = A/400$.**


When non-equivalent nuclei need to be introduced into the second partner the full MFE curve can no longer be calculated analytically, and only the regions of the level crossing lines can be described assuming compactness of the ESR structure of the second partner. Figures 4 and 5 show these regions for a pair that has two spin-$\frac{1}{2}$ nuclei with different HFC constants, equal to *A/10* and *A/40*, in the second partner. Again the familiar "doublet of doublets" with the expected splittings

pattern is clearly seen in both figures. More busy spectra for systems with a more complicated hyperfine structure could have been readily generated, but they bring no new insight and would hardly ever be obtained in experiment, and thus are not included here.

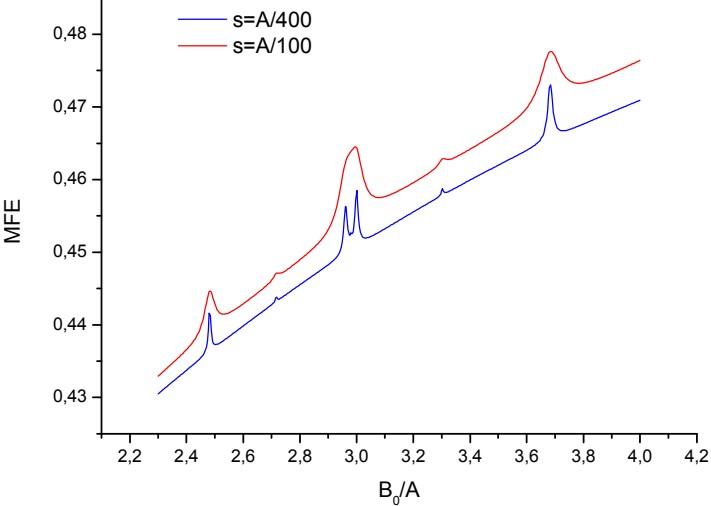

 **Figure 6: Region in the vicinity of** $B_0 = 3A$ **for a pair with 6 equivalent spin-**$\frac{1}{2}$ **nuclei with HFC constant** *A* **in one partner and 2 equivalent spin-**$\frac{1}{2}$ **nuclei with relatively large HFC constant** *A/5* **in the second partner, recombination parameter** *s = A/100*, *s = A/400*.

Finally, Figure 6 shows what happens if the smaller HFC constant becomes not that small and the linearizing
assumptions of this work are pushed too far. The figure, which was obtained by analytic calculation of the full MFE curve,
illustrates the region in the vicinity of the level crossing line at triple HFC constant for a pair that has two spin-$\frac{1}{2}$ nuclei with
HFC constant *A/5* in the second partner. The 1-2-1 pattern becomes distorted, the lines are no longer equidistant, and the
spectrum for a longer-lived pair demonstrates that the central line of the triplet is split. All these features are of course
familiar from conventional second-order ESR spectra and are due to violation of the high-field assumptions. It can be
reasonably claimed that to stay within the linearized paradigm of this work the upper limit for the HFC constants in the
second partner is about one tenth of the HFC constant of the driving partner. Given the couplings in the actual available
experimental systems of 13.7 mT (hexafluorobenzene radical anion) and 15.1 mT (octafluorocyclobutane radical anion),
there is  hope in resolving couplings of the order of milliTesla, which is quite typical for organic radical ions.


## 8 Conclusions

In this work we have provided a full justification for the term "MARY ESR" introduced in (Tadjikov et al., 1996) by showing that under the claimed conditions the level crossing lines will indeed recover an arbitrary ESR spectrum without limitation to the simple cases discussed originally in (Tadjikov et al., 1996). We also hope that the discussed parallels between level crossing spectroscopy and conventional magnetic resonance spectroscopy can help bridge the existing conceptual and perceptional gap between the two fields. Although the discussion relied on the properties of a specific class of systems, radiation-induced radical ion pairs in nonpolar solutions, it may well be that similar approaches could be more easily realized on other correlated spin systems. Given that the language of level (anti)crossings also becomes a unifying language in hyperpolarized magnetic resonance (Sosnovsky et al. 2016), the suggested approaches may come more natural to experts in spin chemistry and magnetic resonance today than they were 20 years ago, and thus may be more useful now rather than alien as they looked originally.

On the more sober side, though, it is clear that many real experimental systems will be more complicated than discussed here. In particular this will be true for photoinduced radical pairs, for which pair partners often cannot be treated as independent electron spins, and additional electron spin-spin interactions like dipolar and exchange must be accounted for. Furthermore, the longer lifetimes of the pairs one is often interested in bring such factors as relaxation and chemical reactivity of the radicals into picture, which also complicates the matters considerably. These factors have received significant attention in the context of the level crossing line in zero field, related to tentative magnetoreception (see, e.g., Efimova and Hore 2008, 2009; Lau et al. 2010; Kattnig et al. 2016a, 2016b; Worster et al. 2016; Kattnig and Hore, 2017; Keens et al. 2018; Babcock and Kattnig 2020), and so far the feeling is that their due account is anything but "simple". Additional interactions destroy the neat partitioning of state space into manageable subspaces similar to introduction of additional nuclei in the "crossing vs anticrossing" section above, and relaxation further adds to this complexity. There is no reason to expect that things will become much easier when moving from zero field crossings to level crossing lines in non-zero fields, and probably comparable effort would be needed to analyze the consequences and implications of such additional complications. The more valuable then seem the simple and comprehensible insights elaborated in this work for a more sterile but still realistic model of a radiation-induced radical ion pair.

**Author contribution**

DS and YM conceived the study. VB derived the key result. DS and VB prepared the manuscript with contributions from all authors.

**Conflict of interests**

The authors declare that they have no conflict of interest.

**Acknowledgements**

The authors are grateful to RSF, project No. 20-63-46034, for financial support, and to paper (Kaptein, 1971) for the inspiration for this work.

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
