# Peer review of "Simple rules for resolved level crossing spectra in magnetic field effects on reaction yields"

_Magnetic Resonance, 2021_

## Author Response (AR1)

In the paper very interesting and useful results on the detection of the EPR spectra of radicals by the MARY method are obtained. The idea is very similar to the transfer of a low frequency audio signal by means of radio carrier frequency modulation. The "carrier frequency" is the intersection of levels in a high field, caused by a radical with large hfi coupling constants. A detailed analysis of the position and width of the resonance caused by the level crossing is carried out. The hyperfine structure of the second radical of the radical pair acts as a "sound signal". The authors show that the level crossing in the zero field does not allow detecting the EPR signal of the second partner, which is similar to that the low-frequency audio signal is not transmitted directly, that is, without modulation by the carrier frequency. The authors carried out a detailed theoretical analysis of the detection of the EPR spectrum of the second partner for various configurations of the hfi spectrum of the first partner (with large hfi constants). The rules are formulated concerning this detection method. I consider the article very useful for the field of magnetic resonance (ESR)  and recommend it for publication in the journal as it is. There are the following notes:

*Author's response:*

*Thank you for a very interesting view at our work. Your interpretation of the behaviour of the spin system of a radical pair in the vicinity of level crossing points that we analysed as a sort of frequency shift similar to FM is really enlightening and  triggers quite some further ideas and analogies. Especially valuable for us is the noted analogy to distinction between the zero-field and non-zero field crossings. It would be also interesting to ponder that what we have here is not only a spectrum shift, but also a pitch shift, or spectrum stretching, due to varying intersection angles of the crossing levels.. In radiotechnics pitch shifting is a time-domain rather then frequency-domain transform, realizable by picking samples faster/slower than they were originally taken, and it is really interesting to ponder what the implications of this might be for a spin system. The link to the field of radiofrequency techniques is indeed quite unexpected and potentially very fruitful as leading outside the box. Thank you very much for your input again. Regarding two your more specific notes, they are certainly valid and benefitial for the readers, and we shall incorporate suitable straightforward amendments for both of them to the revised version of this manuscript.*

More specific comments:

1. It is written: "Laplace transform of singlet state population as a function of applied static magnetic field ... "- it is necessary to clarify that the Laplace transform is done in time domain, otherwise it can be understood that the Laplace transform is done in the magnetic field domain.

*Author's response:*

*Yes, thank you for the suggested clarification. Lines 52-53 in the revised manuscript have been reworded as "theoretical counterpart to experimental observable is the Laplace transform of singlet state population $\rho_{ss}$, taken in time domain, as a function of applied static magnetic field"*

2. It is written: "For the outersmost blocks with \Sigma = + -I... Real outmost states are with \Sigma = + -(I+1). Of course, these states are tripet ones and they does't take part in spin dynamics since they are eigen states but may be it is worth to mention this (?).

*Author's response:*

*Yes, this is certainly true and needs more care. The following passage has been added as Lines 71-73 in the revised manuscript: "The states with maximum possible   i.e., electron spin-triplet states with maximum nuclear spin projection, are isolated eigenstates and are completely excluded from pair spin evolution. For the outersmost blocks involved into spin evolution with   there are only three states with eigenvalues"*

Anonymous Referee #2

The simplest model for analysing the spin dynamics of radical pairs is the so-called 'exponential model' and the simplest description of radical pair spin-state evolution is under the action of a spin Hamiltonian that contains only electron Zeeman and hyperfine coupling terms for each of the individual radical pair partners. This description of radical pairs has long been used to provide useful insights into the behaviours of radical pairs and in particular is often the starting point for simulation of dependence of magnetic field effects on applied magnetic field strength - the so-called MARY curve. In this work the authors take a new analytical approach to deconstructing the predictions of this model. This results in a set of rules that provide very useful insights into both the key features of MARY curves, but that also predict the ability to use MARY as a spectroscopic method yielding spectra effectively equivalent to conventional ESR spectra under particular constraints of the relative hyperfine structure of the pair partners.

I believe that the analytical approach is an extremely useful one, with important predictive power. In particular, I found that the insight it provides over the essential 'canceling' of crossings and anti-crossings and the resulting impact on, for example, the so-called 'low-field effect (LFE)' highly enlightening. In this particular case (and indeed perhaps more so in the other cases discussed) the rules and analytical framework might lead to important new usage cases for 'MARY spectroscopy.' The authors do a good job of relating this approach back to existing ones and highlighting the new insights as well as confirming that it leads to the same existing predictions and understanding.

Therefore, I have no hesitation in recommending that this paper be published as is. However, I do have some observations that the authors may wish to consider (i.e. these are fully optional but may be worth giving some thought to).

*Author's response:*

*Yes, thank you for your thoughtful analysis of our contribution and placing it into the right context. Your appreciation of the value of simple analytical approach as used by us here is very rewarding for us, and we are happy that colleagues find it as useful as we do. There is little else we can add to you summary of our work given in the introductory part of your comment. However, especially valuable also were your specific comments as follows:*

1) Accessibility

The paper is clearly written and while it contains a large number of equations, it is quite straightforward and logical to follow. However, I think the lack of any visual representation of the findings is to some extent a missed opportunity. I feel that in the key example domains the authors consider, it might be useful to provide example spectra and highlight the features that the interpretation is pointing at. Personally I find some problems easier to think about mathematically and others visually. By providing an additional visual representation, it may increase the accessibility of the findings to a broader audience.

*Author's response:*

*Yes, this is certainly true. We took you suggestion seriously and have added a new section to the manuscript, "7 Compendium of typical resolved spectra", showing and discussing typical MARY-ESR spectra and their relation to conventional ESR. Thank you for this suggestion, we hope the readers will appreciate it.*

2) Extensions

As I indicated at the beginning of this comment, the rules are based on a very simple model of RP reactions which does not account for electron exchange or dipolar interactions, incoherent spin relaxation or spin-selective reaction (to name a few). Given the importance of some of these factors, particularly in relation to some of the realistic example systems provided, I wonder if the authors could say even a little about how some of these factors might influence the simple predictions? This may also be useful to experimentalists in thinking about real systems that can exploit some of the predictions provided.

*Author's response:*

*Yes, this is certainly also true. However, this turned out to be not so easy to accomplish. As an attempt to accommodate your suggestion we have added a new paragraph to the Conclusions*

*section and 9 new references to prior works that deal with these issues where they have been elaborated, for the level crossing line in zero field. The paragraph is as follows:*

*"On the more sober side, though, it is clear that many real experimental systems will be more complicated than discussed here. In particular this will be true for photoinduced radical pairs, for which pair partners often cannot be treated as independent electron spins, and additional electron spin-spin interactions like dipolar and exchange must be accounted for. Furthermoe, the longer lifetimes of the pairs one is often interested in bring such factors as relaxation and chemical reactivity of the radicals into picture, which also complicates the matters considerably. These factors have received significant attention in the context of the level crossing line in zero field, related to tentative magnetoreception (see, e.g., Efimova and Hore 2008, 2009; Lau et al. 2010; Kattnig et al. 2016a, 2016b; Worster et al. 2016; Kattnig and Hore, 2017; Keens et al. 2018; Babcock and Kattnig 2020), and so far the feeling is that their due account is anything but "simple". Additional interactions destroy the neat partitioning of state space into manageable subspaces similar to introduction of additional nuclei in the "crossing vs anticrossing" section above, and relaxation further adds to this complexity. There is no reason to expect that things will become much easier when moving from zero field crossings to level crossing lines in non-zero fields, and probably comparable effort would be needed to analyze the consequences and implications of such additional complications. The more valuable then seem the simple and comprehensible insights elaborated in this work for a more sterile but still realistic model of a radiation-induced radical ion pair."*

*I'm afraid there is not much we can currently do beyond these general words, these are serious issues which require dedicated stidues.*

3) Small errors.

I provide no adjustment to grammatical errors in the manuscript, but I did notice a small number of typographical errors that either directly impact the scientific meaning, affect technical terms or may confuse. I list these below:

- Line 195 'produce crossings Of Eq. (7)' —> 'produce crossings of Eq. (7)'
- Line 343 'in addition to for equivalent fluorines' —> 'in addition to four equivalent fluorines'
- Line 354 'donor-acceptor diads' —> 'donor-acceptor dyads'

*Author's response:*

*Yes, thank you very much for catching these, we have corrected them all, as well as several other minor things noted upon second reading*

Additional changes to the revised manuscript:

We have slightly changed the passage around expression 40 (Line 415) following discussion with the editorial staff to avoid color in formulas: the expression has been split in two and color coding was removed.